# Nanoporous Silica Entrapped Lipid-Drug Complexes for the Solubilization and Absorption Enhancement of Poorly Soluble Drugs

**DOI:** 10.3390/pharmaceutics13010063

**Published:** 2021-01-06

**Authors:** Hey-Won Shin, Joo-Eun Kim, Young-Joon Park

**Affiliations:** 1College of Pharmacy, Ajou University, Worldcup-ro 206, Yeongtong-gu, Suwon-si 16499, Korea; shw123n@nate.com; 2Department of Pharmaceutical Engineering, Catholic University of Daegu, Hayang-Ro 13-13, Gyeongsan City 38430, Korea; 77jooeun@naver.com

**Keywords:** dutasteride, nanoporous silica entrapped lipid-drug complex, lipid formulation, solid dosage form, nanoemulsion

## Abstract

This study aims to examine the contribution of nanoporous silica entrapped lipid-drug complexes (NSCs) in improving the solubility and bioavailability of dutasteride (DUT). An NSC was loaded with DUT (dissolved in lipids) and dispersed at a nanoscale level using an entrapment technique. NSC microemulsion formation was confirmed using a ternary phase diagram, while the presence of DUT and lipid entrapment in NSC was confirmed using scanning electron microscopy. Differential scanning calorimetry and X-ray diffraction revealed the amorphous properties of NSC. The prepared all NSC had excellent flowability and enhanced DUT solubility but showed no significant difference in drug content homogeneity. An increase in the lipid content of NSC led to an increase in the DUT solubility. Further the NSC were formulated as tablets using D-α tocopheryl polyethylene glycol 1000 succinate, glyceryl caprylate/caprate, and Neusilin^®^. The NSC tablets showed a high dissolution rate of 99.6% at 30 min. Furthermore, NSC stored for 4 weeks at 60 °C was stable during dissolution testing. Pharmacokinetic studies performed in beagle dogs revealed enhanced DUT bioavailability when administered as NSC tablets. NSC can be used as a platform to develop methods to overcome the technical and commercial limitations of lipid-based preparations of poorly soluble drugs.

## 1. Introduction

The oral route is the preferred route of drug administration for approximately 62% of all commercially available pharmaceutical products [1]. Approximately 40% of pharmaceutical products in the market are of poorly water-soluble drugs [2,3]; therefore, many studies have been performed to improve the solubilization of drugs through techniques, such as size reduction, salt formation, solid dispersion, hot-melt extrusion, or formation of salt, prodrugs, and co-crystals. Alternatively, the utilization of self-nanoemulsifying drug delivery system (SNEDDS) self-microemulsifying drug delivery system (SMEDDS), nanoemulsions and micelles has also been reported [4,5,6,7,8].

The bioavailability of oral dosage forms is affected by various factors, such as aqueous solubility, drug permeability, dissolution rate, and susceptibility to efflux mechanisms [9,10]. As drug solubility is a limiting factor in the oral absorption of BCS Class II drugs such as dutasteride (DUT), it is necessary to increase their solubility in gastrointestinal fluids for enhanced bioavailability [10,11,12,13,14]. DUT is a competitive inhibitor of type I and type II 5-α-reductase and has been globally approved under the brand name Avodart^®^ soft capsule (GlaxoSmithKline) for the treatment of benign prostatic hyperplasia and androgenetic alopecia [15,16,17]. In addition, DUT is a lipophilic model drug, with a logP of 5.6 and aqueous solubility as low as 0.038 ng/mL [18,19].

Avodart^®^ is formulated as a liquid-filled gelatin capsule with lipid such as glyceryl caprylate/caprate to improve the solubility of DUT. Lipid-based formulations and drug delivery systems, such as emulsions, micellar systems, liposomes, niosomes, and self-emulsifying micro emulsions, enhance the permeability of lipophilic drugs (logP > 5) [20,21,22,23,24]. Soft gelatin capsules are the most common lipid-based dosage forms [25,26].

It has been previously reported that the solubilization of lipophilic drugs such as dutasteride is the crucial for oral absorption; liquid lipids, such as lauroglycol, D-α tocopheryl polyethylene glycol 1000 succinate, glyceryl caprylate/caprate, propylene glycol monolaurate, glycerol monocaprylate, and glyceryl tricaprylate/caprate, show the best solubility enhancement among all solubilizing agents. Formulations containing liquid lipid are mainly administered as soft capsules due to pharmaceutical limitation.

Soft capsules require special manufacturing equipment and facilities and are associated with higher manufacturing and startup cost and fewer outsourcing options than for tablet manufacturing. In addition, few companies have expertise in soft encapsulation and synergistic lipid formulations. Moreover, there is a limit on the number of excipients that can be used in the formulating soft capsules owing to the stability of gelatin itself and gelatin–additives compatibility [27,28,29,30,31].

In general, the use of oil-based solubilizers in tablet formulations causes stability problems. Hence, lipophilic drugs are mostly manufactured as soft capsules; however, their low stability leads to leakage of the liquid oil component thus limiting their use. Therefore, a system is needed for the entrapment of oil containing drug formulations and to formulate a solid oral dosage form, mostly preferred by patients. Previous studies have demonstrated the use of common adsorbents to prevent liquid oil leakage, but with certain limitations [32,33,34]. Although the development of tablets using oil and surfactants has been extensively studied, the development of stable tablets using liquid oils and surfactants has been challenging.

In this regard, mesoporous silica provides an alternative for enhancing the solubility and bioavailability of poorly water-soluble drugs. There are various technologies to solubilize poorly soluble drugs, and many studies have been attempted to prepare a solid dosage form by using a solubilization technology using mesoporous silica. These technologies include solid dispersion, emulsion, multi-coating, wetness impregnation method, supercritical carbon dioxide, rotary evaporation, and freeze-drying method [35,36,37,38,39,40]. Mesoporous silica has a pore size of 2–50 nm, with a large pore volume, large surface area, and adjustable particle size [41,42,43]. Since most of these manufacturing methods do surface adsorption, the loading efficiency is low, and the problem of encapsulating and stabilizing must be overcome. Therefore, the NSC system we studied aims to continuously maintain the solubilization system by entrapping lipids and APIs in silica nanopores, increasing the encapsulation efficiency by high, and stabilizing them.

It has been used as a drug delivery system to improve the solubility and increase the bioavailability of poorly water-soluble drugs [44,45,46,47,48,49,50,51,52]. In this study, a nanoporous silica entrapped lipid-drug complex (NSC) was prepared to maximize the oral absorption of the poorly water-soluble drug, DUT. The physicochemical properties and solubilizing ability of NSC, a complex comprising a nanoporous silica carrier and lipid mixture, were evaluated to improve the bioavailability of DUT. NSC was formulated as a solid dosage form; the formulation stability, dissolution profiles, and pharmacokinetic (PK) profile of the NSC tablets were investigated. The possibility of NSC as a new solubilization system can be verified based on the convenience of its manufacturing process. In this study, we aimed to propose a commercially applicable and stable solid dosage form for oral administration with higher dissolution and bioavailability than those of other lipid formulations.

## 2. Materials and Methods

### 2.1. Materials

Dutasteride, with particle size of 1.73 µm (D50) and 4.48 µm (D90), was purchased from Dr. Reddy’s Laboratories (Hyderabad, Telangana, India). Magnesium aluminometasilicate (Mg/Al-SiO_2_, Neusilin^®^ US2 (Neu)) was purchased from Fuji Chemical Industries Co., Ltd. (Gohkakizawa, Toyoma, Japan). D-α Tocopheryl polyethylene glycol 1000 succinate (TPGS), glyceryl caprylate/caprate (GCC), and propylene glycol monolaurate (PGL) were obtained from Merck KGaA (Darmstadt, Germany). Croscarmellose sodium (CCS), microcrystalline cellulose (MCC), low-substituted hydroxypropyl cellulose (L-HPC), sodium stearyl fumarate (SSF), and silicified microcrystalline cellulose (SMCC) were obtained from JRS Pharma Co., Ltd. (Patterson, NY, USA); isomalt (IM) was obtained from BENEO-Palatinit GmbH (Mannheim, Germany). Hydroxypropyl methylcellulose (HPMC)-based coating agent (Opadry^®^ 03B28796 White) was obtained from Colorcon Asia Pacific Pte Ltd. (Suwon, Korea). All other chemicals were of reagent grade and obtained commercially. All excipients for preparing the formulation were of pharmaceutical grade and obtained commercially.

### 2.2. Ternary Phase Diagram

To investigate the concentration range of components for the existing boundary of microemulsions, ternary phase diagrams were constructed using the water titration method. Ternary phase diagrams were prepared with oil (GCC), surfactant (TPGS), and water. The GCC and TPGS mixture was prepared at weight ratios of 0:10, 1:9, 2:8, 3:7, 4:6, 5:5, 6:4, 7:3, 8:2, 9:1, and 10:0. These mixtures were diluted dropwise with distilled water under moderate agitation. The samples were classified as microemulsions when they appeared as clear liquids.

### 2.3. Preparation of NSCs

Briefly, 3.9–11.3 g of TPGS and/or 3.9–6.2 g of GCC or PGL were dissolved in 5 mL ethanol by magnetic stirring at 25 ± 3 °C and 60% relative humidity (RH). DUT (0.2–1.0 g) was added to this solution and stirred at 25 ± 3 °C and 60% RH until a clear solution was obtained. This solution was used as the DUT lipid solution. Nanoporous silica was placed in beakers and agitated with the impeller of a mechanical stirrer (Model ZZ-1000S, EYELA, Kyoto, Japan). The DUT lipid solution was filled in a plastic syringe, fixed to a syringe pump, and then added drop-wise onto the nanoporous silica material (at a rate of 3 mL/min) to incorporate the DUT-lipid complex in the mosoporous silica. Various NSCs (01–07) were prepared by varying the proportion of individual components (Table 1) followed by drying at 60 °C for 5 h. In contrast, a physical mixture-07 was prepared by simply mixing the drug and carrier (1:30, *w*/*w*) (Table 1).

### 2.4. Preparation of NSC Tablets

NSC tablets were prepared using a mixture of NSC, CCS, MCC, L-HPC, SMCC, IM, and SSF (Table 2). They were compressed using a 7 mm round punch in a single punch tablet machine (Autotab-535, Ichihashiseiki, Kyoto, Japan) and then coated with an HPMC-based coating agent such as Opadry^®^ 03B28796 White. The NSC-03 tablet to be used in pharmacokinetic studies was coated with a traditional coating machine (C30, Sejong, Icheon, Korea). In detail, 32 g coating agent was dissolved in 368 g 70% ethanol. Then they were mixed to form the coating solution. A traditional coating pan was utilized, with the tilt angle being 45°. The rotating rate, spray rate of coating solutions, and product temperature were set at 30 rpm, 8 mL/min, and 40 °C, respectively. At this time, 3% per NSC tablet was coated, and then the drying temperature was performed at 45 °C for 5 min.

### 2.5. Physicochemical Properties of NSC

#### 2.5.1. Scanning Electron Microscopy

The surface morphologies of DUT, Neu, and NSC were examined using scanning electron microscopy (SEM; JSM-5200, JEOL, Tokyo, Japan). Specimens were deposited on the double-sided adhesive carbon tape on a working stage and gold plated (150 Å of thickness) before imaging.

#### 2.5.2. Differential Scanning Calorimetry

Differential scanning calorimetry (DSC; DSC250, TA Instruments, New Castle, DE, USA) was performed at a heating rate of 10 °C/min from 40 °C to 280 °C, with nitrogen gas at a flow rate of 50 cc/min. Typically, 3–5 mg of the sample was placed in an aluminum pan and measured.

#### 2.5.3. Powder-X-ray Diffraction

The samples of powder X-ray diffractometer (P-XRD, D/max-2500V/PV, Rigaku, Tokyo, Japan) were exposed to Cu K-α radiation at a scan rate of 2°/min over the 2θ range of 2–40°. Approximately 200–1000 mg of the specimen was placed in the sample holder. The results were obtained as peak intensity versus 2θ.

#### 2.5.4. Flowability

The flowability was determined by measuring the angle of repose (A.R.) and Hausner ratio according to the method of <1174> powder flow in USP Pharmacopeia [53]. Briefly, a glass funnel was fixed by positioning its tip at a fixed height (H) (2.5 cm above a wax paper) placed on a horizontal surface. The NSC powder was poured through a funnel, and the A.R. was measured with a 180° angle ruler using the formula tan (θ) = H/R, where R is the radius of the cone pile. NSC (10 g) was placed in a 25 mL graduated cylinder and the volume and mass were measured to determine the bulk density (B.D. (g/mL)). This mass was tapped 250 times per minute with tap density tester (Tap density, SVM12, ERWEKA, Langen, Germany) according to USP pharmacopeia (method II of <616> bulk density and tapped density of powder) and the mass and volume were remeasured to calculate the tap density (T.D. (g/mL)) [54]. The values of B.D. and T.D. were applied to the Equations (1) and (2) to obtain Carr’s Index and Hausner ratio to evaluate flowability [55,56]:Carr’s Index (%) = 100 × (B.D. − T.D.)/B.D., and (1)
Hausner ratio = T.B./B.D.(2)

#### 2.5.5. Loading Capacity

We determined the loading capacity of the lipid-drug complex in the nanoporus silica by the flowability of NSC powder. When the Carr’s index, which is the factor of flowability, was 30 or more, it was selected as the limit of loading capacity. This is because when it is 30 or more, the content uniformity of tablet is poor, and problems occurs in tablet manufacturing. The determination of loading capacity is as follows. After preparing NSC powders by applying a lipid-drug complex of 70 to 300% relative to the weight of the silica carrier, their bulk density and tapped density were measured to evaluate the Carr’s index.

The amount of lipid complex adsorbed on the surface of NSC and entrapped lipid was distinguished by the following method. In order to measure the ratio of the entrapped lipid-drug complex, its amount was measured by evaluating the amount of DUT in the NSC because the amount of lipid complex and the amount of DUT are proportional. The amount of lipid complex adsorbed to the surface was calculated by subtracting the amount of the entrapped DUT from the total amount of DUT in NSC. To determine the total amount of DUT in the NSC, Samples were taken as 284 mg of NSC powder, and placed in a 25 mL beaker, 70% ACN was added to 10 mL. This solution was stirred at 200 rpm with a magnetic bar for 1 h at 25 ± 3 °C. This solution was filtered through a 0.45 µm PTFE syringe filter, then accurately taken with a 1 mL whole pipette, placed in a 10 mL volume flask, and adjusted to the volume with 70% ACN. This was analyzed according to HPLC Section 2.6.2.

To determine the entrapped amount of DUT in the NSC, Samples were taken as 284 mg of NSC powder, and placed in a 25 mL beaker, 10 mL of 30% EtOH was added and stirred at 200 rpm with a magnetic bar for 5 min at 25 ± 3 °C. The solution was centrifuged at 3000 rpm for 5 min. The above procedure was repeated three times. The supernatant was discarded and added 10 mL of 70% ACN. This solution was stirred at 200 rpm with a magnetic bar for 1 h at 25 ± 3 °C. This solution was filtered through a 0.45 um PTFE syringe filter, then accurately taken with a 1 mL whole pipette, placed in a 10 mL volume flask, and adjusted to the volume with 70% ACN. This was analyzed according to HPLC Section 2.6.2.

#### 2.5.6. Content Homogeneity

Twenty g of prepared NSC and were placed in a 30 mL intermediate bulk container and shaken for 10 min in a container mixer. To measure content homogeneity, each 500 mg of NSC was sampled from the upper, middle left, middle right, lower left, and lower right positions of the intermediate bulk container. The homogeneity of DUT in NSC was evaluated by random sampling of five powder. The content of DUT of NSC obtained from each position of the vessel was analyzed using high-performance liquid chromatography (HPLC), according to the DUT content test method as mentioned Section 2.6.2.

#### 2.5.7. Equilibrium Solubility of DUT in NSC

NSC powder, DUT, and the physical mixture were precisely weighed to obtain 10 mg equivalent of DUT and placed in a 10 mL clear vial. Then, 5 mL of deionized water (DW) was added to the vial and stirred at 800 rpm and 25 ± 0.5 °C using a magnetic stirrer. After stirring for 24 h, the mixture was filtered through a 0.45 µm polytetrafluoroethylene (PTFE) syringe filter. The filtered solution was accurately taken with a 1 mL pipette, placed in a 10 mL volume flask, and diluted with 70% acetonitrile solution. This filtrate was analyzed using HPLC as mentioned in Section 2.6.2 to evaluate the solubility of DUT.

### 2.6. In Vitro Release of NSC Tablets

#### 2.6.1. In Vitro Release

The in vitro release study was conducted in 900 mL of 0.1 N HCl solution containing 0.05% sodium lauryl sulfate (SLS) in a 1000 mL vessel maintained at 37 ± 0.5 °C. The NSC tablets were added to the containers of a type II (paddle method, USP23) dissolution apparatus and operated at 50 ± 0.02 rpm. Aliquots (5 mL) were drawn according to the release times, and the amount of dissolved DUT was analyzed using HPLC, as described in Section 2.6.2. Before injection, the sample was filtered through a 0.45 μm PTFE syringe filter. The injection volume was 100 μL. The disintegration time was measured by observing the time at which the NSC tablet completely disintegrated in the vessel.

#### 2.6.2. HPLC Analysis

The content homogeneity and in vitro release of DUT was analyzed using an Agilent HPLC system, (1260 series, Agilent, Santa Clara, CA, USA) on a phenyl column (Zorbax SB-phenyl, 150 × 4.6 mm, 3.5 μm) equipped with a photodiode-array detector. The mobile phase was acetonitrile:DW (70:30, *v*/*v*), filtered through a nylon 0.45 μm membrane filter. The flow rate was 1 mL/min, temperature of the column was 40 °C, injection volume was 20 μL, and wavelength of the detector was 210 nm. DUT was quantitatively analyzed by preparing a sample solution of 10 µg/mL diluted with mobile phase. The tailing factor was not more than 1.5, and the relative standard deviation (RSD) for replicate injections was not more than 2.0%. To make a standard stock solution, 10 mg of DUT was weighed accurately and dissolved in 10 mL of methanol. The assay samples were prepared by diluting it with the diluent to obtain concentrations between 0.05 and 1 mg/mL. The standard deviation (SD) of precision and accuracy was <2%. The calibration curve was rectilinear with a correlation coefficient of 0.999.

### 2.7. Pharmacokinetics Studies

Twelve male beagle dogs (16–28 months old), weighing 9–13 kg, were obtained from Covance Research Products Inc. (Denver, PA, USA). The 12 male beagle dogs were administered DUT control formulation (Avodart^®^ 0.5 mg capsule; reference drug) and NSC-03 tablets in a crossover study design. The formulations were administered orally, with 30 mL of water. Blood samples, 3 mL per animal, were collected in heparinized tubes 0.5, 1, 2, 3, 4, 6, 24, 48, and 72 h after administering the NSC tablets and control formulation. Plasma was separated by centrifugation (4 °C) at 4000 rpm for 10 min. The plasma sample was frozen and stored at −80 °C until analysis. DUT concentrations in the plasma samples were determined using an Agilent 1290 liquid chromatography with tandem mass spectrometry (LC-MS/MS). All procedures in the protocol complied with the Animal Welfare Act and the Guide for the Care and Use of Laboratory Animals and were approved (IRB No. P173013, Identification code: E2017245, 10 October 2017,) by the ethics committee of KPC Lab (Gwangju, Korea).

### 2.8. Statistical Analysis

The data are expressed as mean ± SD. Comparison of the mean between groups was performed using single factor variance analysis, and the least significant difference test was used for pairwise comparison. Differences were considered statistically significant if the *p* values were ≤0.05. Minitab^®^ software (Version 18, Minitab Inc., University Park, PA, USA) was used for all statistical analyses. Non-compartmental analysis for deriving pharmacokinetic parameters was performed with Phoenix™ WinNonlin software (Ver. 6.4; Pharsight, Mountain View, CA, USA). The area under the curve (AUC) from time 0 to the time for the last measured concentration (AUC_0–t_) was calculated using the linear trapezoidal method. AUC from time 0 to infinity (AUC_0–∞)_ was calculated as the sum of AUC_0–t_ and the ratio of last measured plasma concentration to elimination rate constant. C_max_ was calculated as the maximum measured plasma concentration over a specified period. T_max_ was calculated as the time to reach the maximum measured plasma concentration. The pharmacokinetic parameters of NSC-03 tablets and Avodart^®^ soft capsule formulation were determined. We compared individual C_max_ and AUC values and their ratios (test/reference) using log-transformed data. The means and 90% confidence intervals (CIs) were analyzed using parametric analysis of variance (ANOVA). A two-way ANOVA was used to assess the effects of the formulation, time period, and sequence of administration in crossover design on the pharmacokinetic parameters as fixed effects and the effects of subjects nested within the sequence as a random effect. Other main effects were tested at 5% level of significance against the residual error (mean square error) from the ANOVA model as the error term. Owing to the nature of normal-theory CIs, this was equivalent to performing two one-sided *t*-tests at 5% level of significance.

## 3. Results and Discussion

### 3.1. Ternary Phase Diagram

The microemulsion region for improving solubility of DUT was confirmed by drawing a ternary phase diagram according to the composition ratio of lipid and carrier, using oil component GCC and surfactant TPGS. As shown in Figure 1a, the formation of microemulsion was confirmed at 4:6 to 6:4 composition ratios of TPGS to GCC. The area of formation of translucent microemulsion was the largest when the composition ratio of GCC and TPGS was approximately 5:5 [57,58,59]. Therefore, an optimized lipid complex for microemulsion formation was obtained with a TPGS to GCC was 5:5. The formation of 4:6 to 6:4 composition ratio emulsion was also confirmed through visual observation (Figure 1b). Phase separation began at 7:3, with GCC as the oil component. Therefore, the translucent region where phase separation was not possible, microemulsions composition was in the range 4:6–6:4.

### 3.2. Physicochemical Properties of NSC

#### 3.2.1. Morphology

To confirm the characteristics of drug encapsulation, SEM images of the physical mixture and NSC-03 were acquired and compared (Figure 2). When the physical mixture of Neu and DUT was observed at a 300× magnification, Neu was seen as spherical particles with nano sized pores, and DUT had a cotton-like appearance. The morphology of DUT adsorbed on the NSC surface is shown in Figure 2a. As shown in Figure 2b,c, Neu had a rough surface with fine porosity. On the other hand, the surface of NSC containing lipids was smooth and almost non-porous. In addition, NSC was confirmed to have a closed pore structure. This was because DUT and lipids filled the surface and interior of the silica nanopores. As shown in Figure 2c, (3000× magnification), lipid-containing DUT was entrapped in the pores of NSC. Therefore, it could be concluded that DUT in a solution of lipid and ethanol, when added to a silica carrier, got entrapped into the nanopores of the silica carrier. This method was similar to loading as in the wet-granulation method. It was found that the DUT-lipid complexes were incorporated in the nanopores of silica carrier by the dropping method, but their physical mixtures led to adherence on the silica carrier rather than encapsulation.

#### 3.2.2. Thermodynamic Analysis and Crystallinity

According to the results of DSC thermal analysis, DUT showed endothermic peaks at 167 °C and 250 °C (Figure 3a). However, these characteristic peaks of DUT were not observed for the NSC prepared using solvent dropping method (Figure 3a). In contrast, for the physical mixture of DUT and silica carrier, an endothermic peak was observed at 170 °C. From XRD, the 2θ value of DUT in the physical mixture was 19.118; this was due to the poor incorporation of DUT in the nanoporous silica carrier. As shown in Figure 3b, the crystalline peak corresponding to DUT was not observed in NSC. DSC and XRD data revealed that the drug in NSC was stably incorporated within the nanoporous silica carrier, indicating amorphous nature of DUT in the NSC. As NSC was prepared using wet granulation method with an organic solvent (ethanol) and then dried; the dried product contained lipids and the poorly soluble drug DUT dissolved in the silica carrier. This was possibly the reason why DUT was found to be in an amorphous form in both thermodynamic and crystallinity analyses of the NSC. Therefore, it can be inferred that the NSC is a well-integrated complex of DUT, silica carrier and lipids.

#### 3.2.3. Flowability

Powder flowability is an important factor in tablet production. During tablet manufacturing procedure, using a tablet press machine, powders with low flowability may cause non-uniformity in tablet weight and content. This is due to variations during die cavity filling from the hopper [60]. Therefore, flowability is an important process parameter in assuring content uniformity in tablets. Good flowability is predicted only when the Carr’s index value is ≥10 and ≤30.

As shown in Table 3, the flowability of Neu-based NSC improved as the proportion of silica increased. With the same lipid type, as the ratio of lipid to silica in the NSC increased from 50% to 166%, the Carr index, the flowability index, improved from 21 to 15. The ratio of lipid to silica did not have a significant effect on flowability. Thus, the flowability of Neu-based NSC was not negatively affected when the composition of lipid and silica was 1:1 or higher. Moreover, it was observed that flowability was greatly affected by the type of lipid used in preparing each NSC. NSC-03, -04, -05, and -06, which were prepared with a lipid:carrier ratio of 4:3 (lipid content 133%) or higher, were compared in this study. Best flowability was observed for mixtures containing both TPGS and GCC (NSC-03). The Carr’s index value was poor in NSC prepared using TPGS and PGL, and TPGS alone. In addition, the flowability of NSC-05, which contained both TPGS and PGL (1:1), was 2 times lower than that of NSC-03. NSC-06, which contained TPGS alone, showed the lowest flowability.

As shown in the Table 3, it was found that the flowability of NSC was largely influenced by the type of lipid rather than the ratio of silica and lipid. Therefore, when the lipid composition was 133% relative to silica, the flowability of NSC as a complex was improved as compared with the flowability of the individual components. In addition, lipid type rather than lipid composition and lipid proportion influenced the flowability of NSC. The composition and lipid type also affect the solubility and flowability of poorly soluble drugs, which are important factors in the manufacturing process of tablets.

#### 3.2.4. Loading Capacity

The flowability of NSC powders with the lipid-drug complex of 250% or more relative to the weight of the silica carrier was poor, having 30 and more of Carr’s index. Therefore, the maximum loading capacity of the lipid complex in the silica carrier for tablet formulation was 250% or less based on the weight of the carrier. This proved that the extraction efficiency of the main component compared to the loading amount was 99.8 ± 0.2%. The entrapped efficiency of lipid-drug complex was 92% based on DUT in NSC powder. The surface adsorption was about 8%, and Most of lipid complex were entrapped in NSC powder. Our distinguished point compared with the previous studies is that lipid complex and drug are entrapped more than 90% in nanopore of NSC powder, so it could have a highly efficient entrapment and high stability. Otherwise previous papers were reported that surface-adsorption rates of drug in mesoporous silica complex was 60% and more. [61].

### 3.3. Characterization of NSC and NSC Tablet

#### 3.3.1. Drug Content Homogeneity in NSC

The drug content homogeneity of active pharmaceutical ingredients (APIs) in the dosage form is one of the most critical quality attributes, especially in low-dose products [62,63]. The content homogeneity of DUT in NSCs and physical mixture-07 was assessed as NSCs contained low dose of DUT [DUT:NSC = 1:30 (*w*/*w*)]. As shown in Table 4, DUT content in NSC-03 and NSC-05 are 14.46 ± 0.03 mg/g and 14.43 ± 0.28 mg/g, respectively. DUT content in NSC-06 containing only TPGS is 14.06 ± 0.06 mg/g; physical mixture-07 contains 14.03 ± 0.36 mg/g of DUT (Table 4). TPGS is known to maintain the physical stability of nanosystems and aid the incorporation of hydrophobic drugs into the nanosystem [64]. However, the drug content homogeneity in NSC-03, NSC-05, and NSC-06 was not significantly different from that in physical mixture-07.

The DUT content (mg/g) in NSCs did not differ depending on the location in the container from where the NSC sample was taken. The SD for NSC-03 and NSC-06 was found to be similar (Table 4); the RSD of drug content homogeneity for the different NSCs ranged from 0.21–2.57% (Table 4). For NSC-05 containing PGL, the DUT content homogeneity was found to be poor as indicated by a high SD. This was possibly due to the liquefied state of the lipid (PGL) in the NSC, resulting in poor incorporation and/or decreasing the drug residence time in the pores. The results were similar in the other groups, demonstrating no significant difference between the nanoporous carrier with lipid and the physical mixture.

#### 3.3.2. Solubility

For accessing DUT solubility, the mixing ratio of TPGS and GCC was fixed at 1:1 and the composition ratio of the lipid to silica carrier was 1:1 (NSC-01), 2:3 (NSC-02), 4:3 (NSC-03), and 5:3 (NSC-04). DUT solubility was found to be 5.41 ± 0.3 µg/mL in NSC-01, 6.67 ± 0.6 µg/mL in NSC-02, 7.35 ± 0.8 µg/mL in NSC-03, and 8.19 ± 1.2 µg/mL in NSC-04 (Figure 4). Therefore, DUT solubility in NSCs increased with an increase in the ratio of lipids.

NSCs with a lipid and silica carrier composition ratio of 4:3 were prepared differently for each lipid type to confirm DUT solubility. NSC-03 was prepared by mixing TPGS and GCC, NSC-05 was prepared by mixing TPGS and PGL, and NSC-06 was prepared using TPGS alone. DUT solubility in NSC-03, NSC-05, and NSC-06 was 7.35 ± 0.8, 8.19 ± 1.2, and 6.22 ± 0.9 g/mL, respectively. Therefore, NSC-03, prepared using TPGS and GCC, showed the highest DUT solubility, whereas, NSC-06, prepared using TPGS showed the lowest DUT solubility. DUT solubility in NSC-07 without lipid was 4.27 ± 0.8 µg/mL and in physical mixture-07 prepared using the physical mixing method was 0.035 ± 0.003 µg/mL. The intrinsic solubility of DUT was 0.004 ± 0.0001 µg/mL. DUT solubility improved by approximately 1800 times in NSC-03 (containing lipids TPGS and GCC), about 1000 times in NSC-07 (containing only DUT and silica), and 8 times in the physical mixture-07. This is because the amorphous drug was highly dispersed in the lipid and had high integrity in the silica carrier. NSC-03, which showed the second highest solubility, was selected, also based on the find-ing that there was no difference in the dissolution rates of NSC-03 and NSC-04. Furthermore, DUT solubility in NSC-03 was approximately 1.7 times higher than that in NSC-07 without lipid and approximately 210 times higher than in physical mixture-07.

#### 3.3.3. In Vitro Dissolution of NSC Tablet

For in vitro dissolution studies, the released DUT content from NSC tablets was compared 30 min after the start of dissolution test. NSC tablets were prepared with various composition ratios of lipids and silica carriers and different types of lipids. The hardness of NSC tablets was 7 mm round, in the ranged from approximately 3 to 5 kPa, the friability was less than 0.1%, the porosity of the tablet was 30.4 ± 1.0%, and the disintegration time ranged from 98 to 131 s.

NSC tablet comprising a 1:3 composition ratio of DUT and silica carrier (NSC-03 tablet containing lipid), NSC tablet without lipid (NSC-07 tablet), and physical mixture tablet without lipid (physical mixture-07 tablet), were prepared. The released content of DUT at 30 min from the start of the dissolution test was determined for these tablets. As shown in Figure 5a, the dissolution rate of DUT in NSC-03 tablet (containing lipids) was 99.6 ± 2.9%, in NSC-07 tablet (containing no lipids) was 68.7 ± 2.8% and in physical mixture-07 was 58.4 ± 2.5%. Therefore, it could be inferred that in the presence of lipids, the dissolution rate of DUT in NSC tablets improved by approximately 1.4 to 1.7 times. In addition, the dissolution rate of DUT also increased in the physical mixture (NSC-07) tablet containing no lipids prepared using the solvent dropping method. Therefore, it could be concluded that the dissolution rate of DUT varies according to the manufacturing method.

As shown in Figure 5b, the dissolution rate of DUT was 60.9 ± 0.2% for NSC-01 tablets, with a lipid and carrier mixing ratio of 1:1 (L:C, 1:1), 78.9 ± 1.9% for NSC-02 tablets (L:C, 2:3), 99.6 ± 2.9% for NSC-03 tablets (L:C, 4:3), and 98.7 ± 1.6% for NSC-04 (L:C, 5:3) tablets. The dissolution rate was the highest for tablets with highest total lipid, and L:C of 4:3 and 5:3, respectively. Thus, for NSC tablets, high lipid composition ratio enhanced the dissolution rate of DUT. The dissolution rate for L:C 5:3 was similar to that for L:C 4: 3, and DUT was completely released from the tablets within 30 min. This was because the solubility of DUT did not increase further with increase in lipid composition. Therefore, it can be concluded that the dissolution of DUT improved when the lipid content was 4 or more in the composition ratio of lipid and silica carriers, with the silica content fixed at 3.

Different NSC tablets were prepared for each lipid type at a lipid and silica carrier composition ratio of 4:3, and the dissolution rates were compared. NSCs with different lipid types were, NSC-03 tablets in which TPGS and GCC were mixed, NSC-05 tablets in which TPGS and PGL were mixed, and NSC-06 tablets using only TPGS. As shown in Figure 5c, the DUT dissolution rate was 99.6 ± 2.9% in NSC-03 containing TPGS + GCC tablets, 88.8 ± 0.1% in NSC-05 containing TPGS + PGL tablets, and 97.8 ± 1.9% in NSC-06 containing TPGS tablets. The dissolution rate was highest in NSC-03 containing TPGS + GCC tablets; it was comparable in TPGS tablets as well than NSC-05 (TPGS + PGL). NSC-05 containing TPGS + PGL tablets showed the lowest dissolution rate. This may be due to lower solubility of PGL, an oil component, as compared to GCC.

DUT is available in the market as Avodart^®^ soft capsule, an original product with the solubilization technology of a lipid-base formulation. In the study by Choi et al. [65], Avodart^®^ was shown to have a dissolution rate of 81.8% at 60 min in a dissolution buffer solution containing 1% *w/v* SLS in 0.1 N HCl. However, in this study, the dissolution of DUT reached 99.6% in 30 min from the NSC tablets, corresponding to complete dissolution for DUT. Therefore, it could be concluded that NSCs showed an increased dissolution rate for DUT as compared to Avodart^®^. Moreover, DUT also showed improved solubility when formulated in NSCs due to enhanced incorporation of DUT. Therefore, NSC as a formulation improves solubility and can be used for developing solid dosage form of tablets with improved dissolution rate.

#### 3.3.4. Stability Evaluation of NSC Tablets via Dissolution Rate

Lipids, such as TPGS, GCC, and PGL, used in NSC are present in semi-solid to liquid form depending on the ambient temperature. Therefore, NSC lipids can be discharged outside the carrier, disrupting the solubilization system for poorly soluble drugs and physically unstable [66,67]. Lipids with high fluidity do not maintain a stable solubilization system. Each NSC tablet with a different lipid type was stored for 4 weeks at a high temperature of 60 °C, and stability was confirmed by the dissolution rate of DUT. This dissolution rate of DUT was compared with the dissolution rate at 30 min from the start of dissolution. TPGS + GCC (NSC-03) tablet with lipid, TPGS + PGL (NSC-05) tablet with lipid, and physical mixture (Physical mixture-07) tablet without lipids were prepared. As shown in Figure 6, the DUT dissolution rate in TPGS + GCC tablets was 99.6 ± 2.9%, and it decreased by approximately 3.4% to 96.2 ± 2.3% after 4 weeks. The DUT dissolution rate in TPGS + PGL tablets was 88.1 ± 0.1%, and it decreased by approximately 13.3% to 76.4 ± 1.1% after 4 weeks. In the physical mixture-07 tablet, the dissolution rate of DUT was 58.4 ± 2.5%, and after 4 weeks, it decreased by 3.1% to 56.6 ± 2.0%.

In the tablets prepared by physical mixing method and by mixing TPGS + GCC, the dissolution rate of DUT decreased but by a small percentage. However, the dissolution rate of DUT in the TPGS + PGL tablet reduced by a higher value than in TPGS + GCC. This might be because of high fluidity of PGL that maintains a liquid state even at sub-zero. The lipid in the silica carrier was not stable, and the incorporation of DUT was poor in that system. Therefore, it was found that the dissolution rate of DUT reduced owing to the collapse of the solubilization system under temperature stress. Nevertheless, the tablets were found to have excellent stability when mixed with TPGS and GCC in NSC. Therefore, it was confirmed that NSC containing TPGS + GCC is a well-integrated form of DUT, lipid, and silica carrier.

### 3.4. Pharmacokinetic Study of NSC Tablets in Beagle Dogs

The pharmacokinetic profile of optimal NSC-03 tablets was compared with that of the original Avodart^®^ soft capsule. Figure 7 presents the plasma concentration of DUT, with time, in beagle dogs. The average of each pharmacokinetic parameter is shown in Table 5. The AUC_t,_ C_max,_ and T_max_ of NSC tablets and Avodart^®^ were 2940.3 ± 1036.2 ng × h/mL and 2843.7 ± 1106.9 ng × h/mL, 67.60 ± 16.37 ng/mL and 63.73 ± 23.75 ng/mL, and 1.79 ± 1.50 h and 1.75 ± 2.03 h, respectively. The 90% CIs for the logarithmic differences in AUC and C_max_ were log 0.86–1.04 and 0.81–1.00, respectively, and the relative bioavailability of NSC tablets to Avodart^®^ was 103%.

DUT is poorly soluble in water (0.0004 μg/mL) and classified as a BCS class II compound; its absolute bioavailability is 40% or less. However, Avodart^®^ soft capsule, the commercially available product is a formulation with improved bioavailability and solubility compared to that of DUT, and its oral bioavailability is known to be approximately 94% [16]. Previously, many studies have tried to convert soft capsules into tablets for enhancing patient compliance [68,69,70,71]. However, the relative BA of general tablet formulation compared with Avodart^®^ soft capsule has been reported to be less than 90%. Michael et al. reported that DUT tablets and hard capsules showed relatively low bioavailability compared with Avodart^®^ soft capsule, being 76% and 73%, respectively. They focused their research on improving the solubility of DUT, improving absorption and making a tablet with relatively similar BA to Avodart^®^ capsule. There is a limit to the production of tablets with equivalent BA to launched soft capsule as the known solubilization technology [72].

In summary, the NSC-03 tablets showed higher DUT solubility than general tablets, and pharmacokinetic parameters indicated increased bioavailability of 103% compared with that of Avodart^®^ soft capsules. This corresponds to approximately a 1.4-fold increase in bioavailability compared with that of the general formulation, and the results were consistent with those of Avodart^®^ soft capsules. Therefore, even if the solubility of DUT increases further, the bioavailability is not expected to increase further. This was also confirmed in a recent study by Min et al. [60]. While, Avodart^®^ soft capsule is an optimal oral formulation with improved solubility and bioavailability, the NSC tablet developed in this study, with improved solubility and bioavailability, is a solid dosage form showing comparable properties to Avodart^®^ soft capsule [73,74]. Despite being a solid dosage form, NSC tablets showed good dissolution and improved bioavailability, similar to Avodart^®^ soft capsules. In addition, the NSC we studied was a complex with a silica carrier prepared by dissolving a poorly water-soluble drug in an oil-based substance, and it could be prepared as a tablet. Moreover, it is an improved formulation compared with general tablets. The pharmacokinetics results revealed that the bioavailability of NSC relative to that of Avodart^®^ soft capsule was 103%. Accordingly, the NSC system could be applied to other lipophilic drugs, such as isotretinoin, tretinoin, and calcitriol, and the possibility of manufacturing a combination tablet of tamsulosin and DUT could also be explored [75,76]. As a major problem associated with formulating combination drugs is the large formulation size when manufactured as general tablets, NSC tablets could be manufactured to ensure patient compliance by overcoming the tablet size issue thus highlighting the potential application value of NSC.

## 4. Conclusions

In this study, a complex system comprising a mixture of DUT-dissolved in lipids and encapsulated in a mesoporous silica particle NSC was used to improve the solubility of DUT using a minimal amount of lipid. The NSC was formulated as a tablet capable of producing a spontaneous emulsion. As a novelty difference of this article, we have proved that the extraction efficiency of the main component compared to the loading amount was 99.8 ± 0.2%. The entrapped efficiency of lipid-drug complex was 92% based on DUT in NSC powder. The surface adsorption was about 8%, and Most of lipid complex were entrapped in NSC powder. Our distinguished point compared with the previous studies is that lipid complex and drug are entrapped more than 90% in nanopore of NSC powder, so it could have a highly efficient entrapment and high stability. Otherwise previous papers were reported that surface-adsorption rates of drug in mesoporous silica complex was 60% and more. We used the dropping method to complex and entrap lipids and DUTs with nanoporous silica carriers. Compared to other studies our NSC was encapsulated and had a high loading efficiency of over 90%. Moreover, it overcomes the problem of recrystallization by separating drugs and lipids and keeps the solubilization system persistent and stable. The solubility of the poorly soluble drug DUT was optimized in the solid dosage form, and the dissolution rate and bioavailability were improved. The NSC, developed in this study, thus provide a commercially applicable solubilization technology platform. The best flowability was observed for the NSC with total lipid and silica carrier composition of 4:3. NSC-03 tablet in our study showed excellent incorporation, good flowability, and good dissolution rate of 99.6%. In addition, this resulted in the same results in the solubility of the DUT of NSC and dissolution rate of the DUT of NSC tablet. The solubility of DUT increased with increase in lipid content. As DUT was completely dissolved in solid dosage forms made from lipids, NSC tablets could be used as stable oral dosage forms with improved dissolution rates and bioavailability. Moreover, as lipid-based formulations of lipophilic drugs, such as soft capsules, are expensive to manufacture, the findings from this study offer solutions to overcome disadvantages of limited knowledge of commercial techniques and manufacturing equipment, associated with formulating with soft capsule manufacturing. Therefore, through our study, we propose an alternative technique to manufacture solid dosage forms that can improve the solubility and bioavailability of lipophilic drugs. It was established that the NSC system incorporating the solubilized lipid component was able to solve problems of leaking and stability reported in previous studies. In addition, the system demonstrated improved dissolution and bioavailability of DUT. Therefore, the system reported in this study can be used for solubilization of other lipophilic drugs. Also, an attempt to formulate the NSC into a tablet, having excellent solubilization ability, was successfully achieved. In addition, this technology promises patient compliance through tablet formulation, which can be orally administered, and therefore have a commercial impact.

## Figures and Tables

**Figure 1 pharmaceutics-13-00063-f001:**
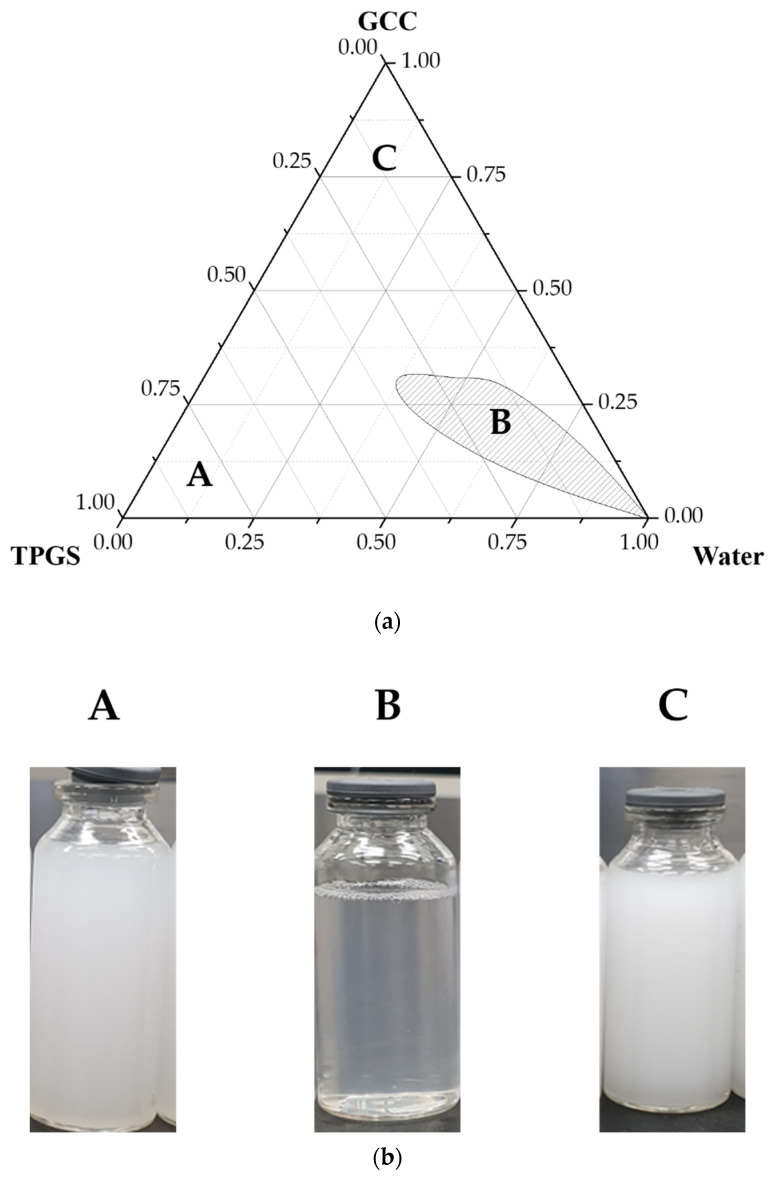
(**a**) Ternary phase diagram of emulsions with glyceryl caprylate/caprate, D-α tocopheryl polyethylene glycol 1000 succinate as oil component, and water, (**b**) image of the emulsions prepared by varying the composition ratio of TPGS and GCC.

**Figure 2 pharmaceutics-13-00063-f002:**
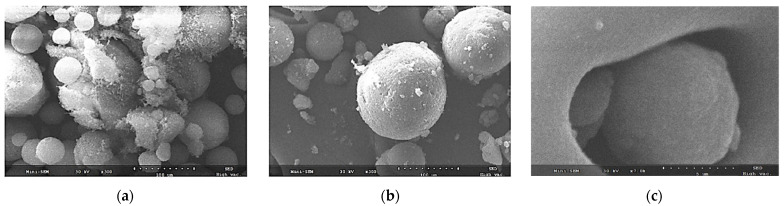
Scanning electron microscopy (SEM) micrographs showing the morphology of (**a**) physical mixture at 300*×* magnification; (**b**) NSC-03 at 300*×* magnification; and (**c**) NSC-03 at 7000*×* magnification.

**Figure 3 pharmaceutics-13-00063-f003:**
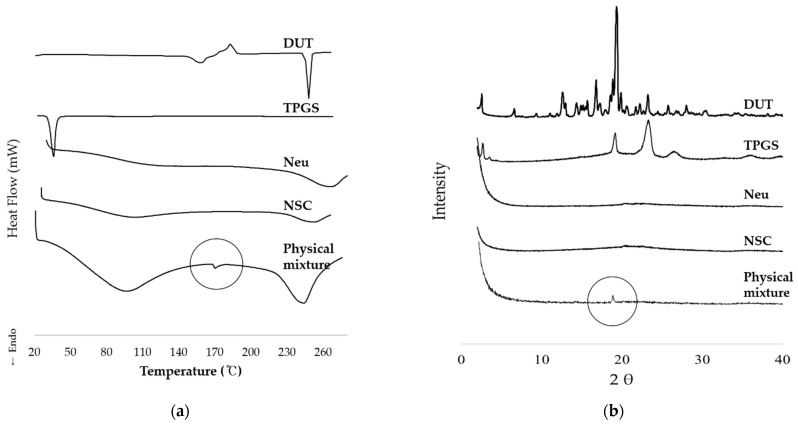
Physicochemical characterization of the interaction between DUT and nanoporous silica entrapped lipid-drug complex. (**a**) DSC analysis; (**b**) HP-XRD analysis. Abbreviations: DUT, dutasteride; DSC, Differential scanning calorimetry; HP-XRD, high powder X-ray diffractometer.

**Figure 4 pharmaceutics-13-00063-f004:**
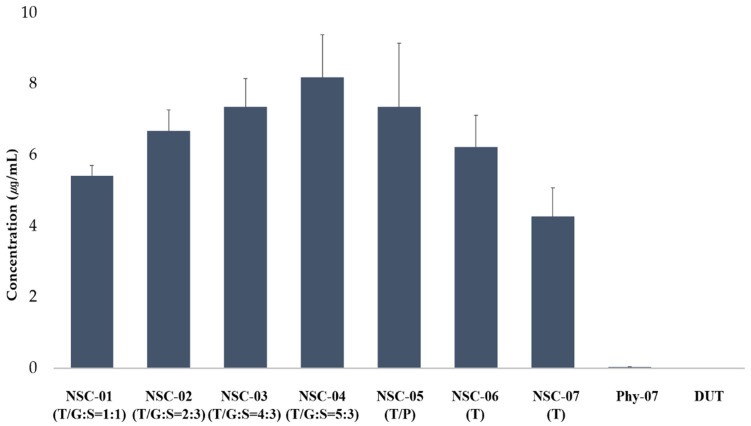
The solubility of dutasteride in NSCs by lipid and silica carrier composition ratio, and in NSC by lipid composition types (*n* = 3). The solubility of dutasteride in physical mixture of DUT and silica carrier, and pure DUT is also shown. Abbreviations: DUT, dutasteride; NSC, nanoporous silica entrapped lipid-drug complex; Phy-07, Physical mixtre-7.

**Figure 5 pharmaceutics-13-00063-f005:**
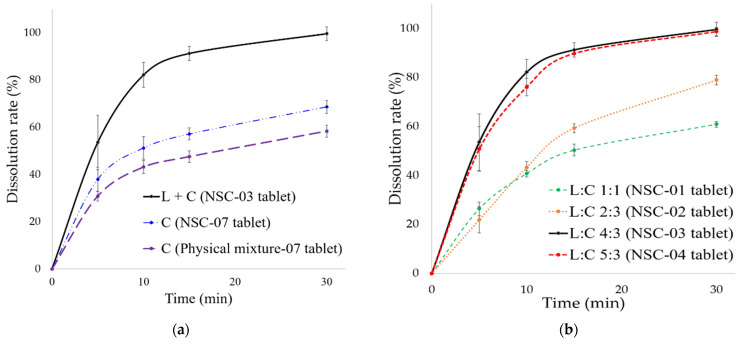
In vitro dissolution of NSC tablets. (*n* = 3) (**a**) NSC tablet with lipids, NSC tablet without lipids, and physical mixture tablet; (**b**) NSC tablet by lipid and silica carrier composition ratio; and (**c**) NSC tablet by lipid component types. Abbreviations: L, lipid; C, carrier; L:C, lipid:carrier; NSC, nanoporous silica entrapped lipid-drug complex; TPGS, D-α tocopheryl polyethylene glycol 1000 succinate; GCC, glyceryl caprylate/caprate; PGL, propylene glycol monolaurate; DUT, dutasteride.

**Figure 6 pharmaceutics-13-00063-f006:**
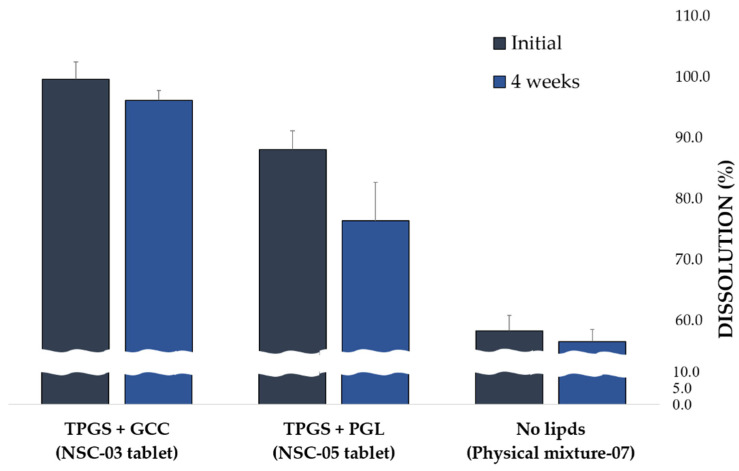
Dissolution rate of dutasteride in different NSC tablets (according to lipid combination) at 30 min from the start of dissolution and after stress test at 60 ± 2 °C for 4 weeks.

**Figure 7 pharmaceutics-13-00063-f007:**
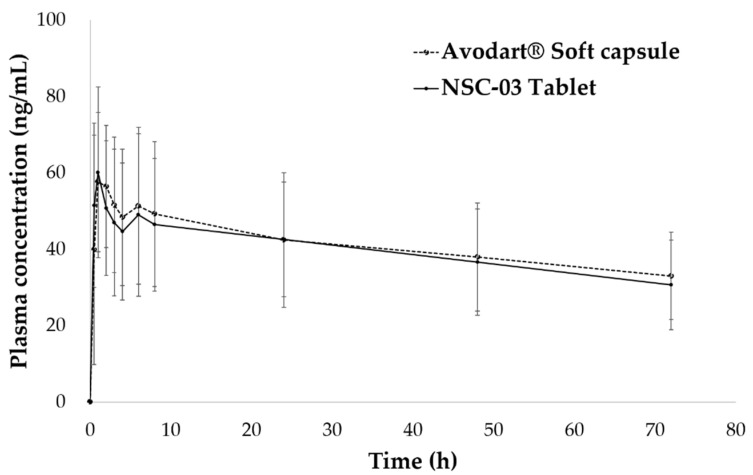
Plasma concentration–time profiles of dutasteride (0.5 mg) after a single oral administration of NSC tablets and Avodart^®^ soft capsules in beagle dogs (*n* = 12).

**Table 1 pharmaceutics-13-00063-t001:** Composition ratio of components in NSCs (%, *w*/*w*).

	DUT	TPGS	GCC	PGL	Carrier *	L:C
NSC-01	1	5	5	-	10	1:1
NSC-02	1	10	10	-	30	2:3
NSC-03	1	20	20	-	30	4:3
NSC-04	1	25	25	-	30	5:3
NSC-05	1	20	-	20	30	4:3
NSC-06	1	40	-	-	30	4:3
NSC-07	1	-	-	-	30	-
Physical mixture-07	1	-	-	-	30	-

* Carrier: Neusilin^®^ US2 (Magnesium aluminometasilicate). Abbreviations: DUT, dutasteride; TPGS, D-α tocopheryl polyethylene glycol 1000 succinate; GCC, glyceryl caprylate/caprate; PGL, propylene glycol monolaurate; L:C, Lipid:Carrier; NSC, nanoporous silica entrapped lipid-drug complex.

**Table 2 pharmaceutics-13-00063-t002:** Composition of NSC tablets (%, *w*/*w*).

	NSC-01 Tablet	NSC-02 Tablet	NSC-03 Tablet	NSC-04 Tablet	NSC-05 Tablet	NSC-06 Tablet	NSC-07 Tablet	Physical Mixture-07 Tablet
NSC-01	7	-	-	-	-	-	-	-
NSC-02	-	15	-	-	-	-	-	-
NSC-03	-	-	21	-	-	-	-	-
NSC-04	-	-	-	18	-	-	-	-
NSC-05	-	-	-	-	21	-	-	-
NSC-06	-	-	-	-	-	21	-	-
NSC-07	-	-	-	-	-	-	10	-
Physical mixture-07	-	-	-	-	-	-	-	10
CCS	9	8	8	7	8	8	9	9
L-HPC	21	21	18	18	18	18	21	21
MCC	21	20	18	18	18	18	21	21
SMCC	21	18	16	16	16	16	19	19
IM	21	18	18	16	18	18	21	21
SSF	1	1	1	1	1	1	1	1
Total weight	150 mg	160 mg	170 mg	220 mg	170 mg	170 mg	150 mg	150 mg

Abbreviations: NSC, nanoporous silica entrapped lipid-drug complex; CCS, Croscarmel-lose sodium; L-HPC, low-substituted hydroxypropyl cellulose; MCC, microcrystalline cellulose; SMCC, silicified microcrystalline cellulose; IM, isomalt; SSF, sodium stearyl fumarate.

**Table 3 pharmaceutics-13-00063-t003:** Flowability of Neu-based NSC.

NSC	A.R. ^1,2^ (°)	Bulk Density ^1^ (g/mL)	Tap Density ^1^ (g/mL)	Carr’s Index ^1^ (%)	Hausner ^1^ Ratio ^1^ (H.R.)
NSC-01	34.1° ± 0.9	0.39 ± 0.00	0.46 ± 0.01	18.1 ± 2.32	1.18 ± 0.02
NSC-02	36.1° ± 1.8	0.38 ± 0.02	0.46 ± 0.01	21.3 ± 2.4	1.21 ± 0.02
NSC-03	33.2° ± 2.9	0.42 ± 0.01	0.48 ± 0.00	14.5 ± 3.7	1.14 ± 0.04
NSC-04	36.7° ± 2.9	0.40 ± 0.01	0.46 ± 0.01	14.9 ± 0.5	1.15 ± 0.00
NSC-05	44.5° ± 1.5	0.42 ± 0.01	0.55 ± 0.01	30.8 ± 2.4	1.31 ± 0.02
NSC-06	56.1° ± 3.1	0.41 ± 0.03	0.57 ± 0.02	36.8 ± 3.8	1.37 ± 0.04
NSC-07	33.1° ± 2.1	0.36 ± 0.01	0.42 ± 0.00	16.9 ± 2.7	1.17 ± 0.03
Physical mixture-07	31.1° ± 0.8	0.33 ± 0.02	0.38 ± 0.01	15.4 ± 3.0	1.15 ± 0.03

^1^ Mean ± S.D. (A.R. *n* = 5, C.I. *n* = 3, H.R. *n* = 3). ^2^ Angle of repose above 40° is considered cohesive.

**Table 4 pharmaceutics-13-00063-t004:** Dutasteride content homogeneity in NSCs.

Assay (Unit: mg/g)	NSC-03 (TPGS + GCC)	NSC-05 (TPGS + PGL)	NSC-06 (TPGS)	Physical Mixture-07 (no Lipid)
Upper	14.47	14.51	13.96	13.52
Middle-Right	14.47	14.26	14.16	13.94
Middle-Left	14.44	14.92	14.06	13.82
Bottom-Right	14.49	14.38	14.06	14.42
Bottom-Left	14.41	14.10	14.07	14.45
Average	14.46	14.43	14.06	14.03
SD	0.03	0.28	0.06	0.36
RSD (%)	0.21	1.94	0.43	2.57
*p* value	<0.05	<0.05	<0.05	>0.05

Abbreviations: SD, standard deviation; RSD, relative standard deviation.

**Table 5 pharmaceutics-13-00063-t005:** Pharmacokinetic parameters of dutasteride after a single oral administration of NSC tablets and Avodart^®^ soft capsules in beagle dogs. All data are expressed as mean ± standard deviation (S D), *n* = 12.

	NSC Tablets	Avodart^®^	90% CI ^4^
AUCt ^1^ (ng·h/mL)	2940.3 ± 1036.2	2843.7 ± 1106.9	0.8641, 1.0454
Relative BA, % (to the Avodart^®^)	103.4%	-	-
C_max_ ^2^ (ng/mL)	67.60 ± 16.37	63.73 ± 23.75	0.8132, 1.0024
T_max_ ^3^ (h)	1.79 ± 1.50	1.75 ± 2.03	-

^1^ AUC_t_: area under the plasma concentration–time curve to the last sampling time. ^2^ C_max_: maximum plasma concentration. ^3^ T_max_: time to C_max_. ^4^ CI: confidence interval.

## Data Availability

The data presented in this study are included in this published article.

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
