# Peer review of "Nanoporous Silica Entrapped Lipid-Drug Complexes for the Solubilization and Absorption Enhancement of Poorly Soluble Drugs"

_pharmaceutics, 2021, doi:10.3390/pharmaceutics13010063_

Round 1

Reviewer 1 Report

In the current manuscript the authors formulated a drug-lipid complex encapsulated in a silica carrier. The results indicate that the solubility-dissolution characteristics of the formulation are superior to that of the drug and the physical mixtures. According to the authors the formulation also has good flow properties. In addition to the above-mentioned improvements, the formulation also behaves better than / equal to that of the capsule when evaluated for PK parameters in an animal model. The article is well written and has a lot of good technical information & could be of potential use to scientists / formulators working with poorly soluble drug molecules and facing issues with manufacturing. That said, there are few issues that need to be ironed out before the work can be deemed acceptable for publication.

  1. Line 59: Porosity is dimensionless. Correct that to pore size.
  2. Section 2.1 mentions a long list of materials. Apart from the lipid, polymer, carrier, and the API where were the other materials used.
  3. Section 2.4: What is the composition of the final tablet?
  4. What is the particle size of the API used?
  5. Section 2.5.4: Why is the tap density measured after 250 taps per minute. Is this according to any compendia?
  6. Did you determine the loading of the complex in the carrier? Were you able to distinguish between the surface adsorbed complex and the entrapped complex?
  7. Line 238: The sentence is incomplete.
  8. Section 3.1: Include pictures of the lipids to demonstrate the translucent nature of different compositions and the rationale behind choosing the design space.
  9. Figure 3: Improve the quality / resolution of the thermograms (DSC) and the diffractograms (XRPD).
  10. Section 3.2.3: Is this the flow data / performance of the NSC or the NSC formulated with MCC and other components?
  11. Could you elaborate on how the solubility experiment was conducted in presence of the NSC?
  12. Was the dissolution performed on NSC or NSC formulated with other excipients?
  13. Line 364-365: The hardness of the tablets is too low to survive coating / handling. What are the dimensions of the tablets used? What is the porosity of the tablets? Again, what is the formulation for these tablets? Were these coated? The materials section mentions OPADRY. Was it used?
  14. Improve the quality of Figure 5. Use colors and symbols to differentiate.
  15. Were the tablets on stability evaluated using DSC and XRPD?
  16. What is the rationale behind using 60C as the temperature for the stability study?

Author Response

Thank you for your comments

Reviewer 2 Report

The Manuscript "Nanoporous silica entrapped lipid-drug complex for solubilization and absorption enhancement of poorly soluble drug" by Shin at el investigate alternative formulation to Avodart, a liquid-filled gelatin capsule of dutasteride. Despite the established rational. The Manuscript was disappointing in terms of English language, presentation, incomplete results/experimental.

Major issues must be resolved before reviewing this manuscript again for publication.

  • Comprehensive English language editing is needed.
  • Many presentation/formatting errors -
  • Results and discussion part is lengthy, text must be rewritten in concise.
  • Table 3 and Figure 4. – Experiments must be performed in triplicates- three different batches/experiments; to perform meaningful statistical analysis. No average nor SD!
  • All figures are of very poor quality
  • Are standard deviations for the dissolution and in vivo studies were as a result of there different samples of the same formulation prepared once? Or from there different formulations prepared differently (three batches)?

Author Response

Thank you for your comments.

Reviewer 3 Report

The manuscript “Nanoporous silica entrapped lipid-drug complex for solubilization and absorption enhancement of poorly soluble drug” deals with the formulation of nanoporous silica entrapped lipid-drug complex (NSC) loaded with dutasteride for improved solubility, and oral absorption. Already many research articles published on porous silica as an adsorbent for lipids. The authors should have discussed the novelty of this study in detail. The methods section and the presentation of results and discussion need to be completely rewritten. I would recommend a major revision before it gets accepted for publication in Pharmaceutics Journal.

Page 1, Abstract, Line 10: Delete ‘A’ before ‘In this study…..’.

Page 1, Abstract, Line 19: Insert the trademark to Neusilin.

Page 1, Introduction, Lines 29-31: The sentence ‘The oral route contributes…..’ is long. Split it into two sentences.

Page 1, Introduction, Line 32: Briefly mention the approaches to improve the solubilization of drugs.

Page 3, Table 1: Mention the carrier name in the table footnotes.

Page 3, Section 2.4: Clearly mention the composition of tablets with a table. This section needs to be improved. Briefly explain the coating method with coating instrument, coating solution composition, and conditions.

Page 3 and 4, Section 2.5.1.: Mention the thickness of gold plating.

Page 4, Section 2.5.3., Line 137: Correct the sentence ‘The samples of high ……’

Page 4, Section 2.5.4, Line 142, and 151: Correct the table number.

Page 4, Table 2: It is very well known and not necessary. Delete the table and cite a reference of USP pharmacopeia.

Page 4, Section 2.5.5.: Mention the quantities of NSC powder and final mixture for NSC taken for content homogeneity.

Page 5, Sections 2.5.6. and 2.6.1: Justify the selection of PVDF and PTFE filters?

Page 6, Section 3.2.1: It is very difficult to analyze the porous nature of the particles with the current images. Include high magnification images with visible surface information. Figure 2C is missing the scale. It is not convincing to accept that the entrapped material is lipid-containing DUT? Needs more studies to confirm that.

Page 7, Section 3.2.2.: Rewrite this section with a discussion about the peaks and changes in their positions.

Page 8, Table 8: Include the standard deviations for bulk density, tapped density, Carr’s index, and Hausner ratio values.

Page 10, Figure 4: Include the error bars.

In conclusion, the results and discussion part is very poorly written and needs to be improved significantly.

Author Response

Thank you for your comments.

Round 2

Reviewer 2 Report

All comments/issue were appropriately addressed ..

Reviewer 3 Report

I appreciate the authors' response to the reviewer's comments and all the changes. Now, I recommend accepting the manuscript for publication.